# Design of a High Sensitivity Pirani Gauge Based on Vanadium Oxide Film for High Vacuum Measurement

**DOI:** 10.3390/s22239275

**Published:** 2022-11-29

**Authors:** Song Guo, Liuhaodong Feng, Shuo Chen, Yucheng Ji, Xinlin Peng, Yang Xu, Yong Yin, Shinan Wang

**Affiliations:** 1School of Microelectronics, Shanghai University, Shanghai 201899, China; 2Shanghai Industrial μTechnology Research Institute, Shanghai 201899, China

**Keywords:** MEMS Pirani gauge, vanadium oxide, high TCR, structure design, fabrication process design, simulation by COMSOL Multiphysics

## Abstract

We have designed a hot-plate-type micro-Pirani vacuum gauge with a simple structure and compatibility with conventional semiconductor fabrication processes. In the Pirani gauge, we used a vanadium oxide (VOx) membrane as the thermosensitive component, taking advantage of the high temperature coefficient of resistance (TCR) of VOx. The TCR value of VOx is $-2{\%K}^{-1}∼-3{\%K}^{-1}$, an order of magnitude higher than those of other thermal-sensitive materials, such as platinum and titanium ($0.3{\%K}^{-1}∼0.4{\%K}^{-1}$). On one hand, we used the high TCR of VOx to increase the Pirani sensitivity. On the other hand, we optimized the floating structure to decrease the thermal conductivity so that the detecting range of the Pirani gauge was extended on the low-pressure end. We carried out simulation experiments on the thermal zone of the Pirani gauge, the width of the cantilever beam, the material and thickness of the supporting layer, the thickness of the thermal layer (VOx), the depth of the cavity, and the shape and size. Finally, we decided on the basic size of the Pirani gauge. The prepared Pirani gauge has a thermal sensitive area of 130 × 130 μm^2^, with a cantilever width of 13 μm, cavity depth of 5 μm, supporting layer thickness of 300 nm, and VOx layer thickness of 110 nm. It has a dynamic range of 10^−1^~10^4^ Pa and a sensitivity of 1.23 V/lgPa. The VOx Pirani was designed using a structure and fabrication process compatible with a VOx-based uncooled infrared microbolometer so that it can be integrated by wafer level. This work contains only our MEMS Pirani gauge device design, preparation process design, and readout circuit design, while the characterization and relevant experimental results will be reported in the future.

## 1. Introduction

A micro-Pirani vacuum gauge is a vacuum pressure monitoring device based on thermal conduction. It realizes pressure detection in a vacuum environment through the Pirani effect and the thermal-electric coupling principle. Some micro-electro-mechanical system (MEMS) devices need to be packaged in a stable vacuum environment to operate, such as MEMS accelerometers and gyroscopes, which have high-speed moving parts that need to be encapsulated in a stable vacuum environment to reduce air damping [1]. In addition, the uncooled microbolometer Focus Plane Arrays (FPAs) are highly thermally sensitive; in order to avoid gas thermal conductance in the package, which will kill the microbolometers’ function, it requires a high vacuum environment [2,3]. These MEMS sensors are widely used in automotive, aerospace, military, and other industries. Therefore, pressure monitoring is an important topic for several industries. Micro-Pirani vacuum gauges are of vital importance in vacuum pressure measurement in modern society. In recent decades, scholars have focused on the development of high-performance micro-Pirani vacuum gauges with a large dynamic vacuum pressure range, a small size, and compatibility with complementary metal oxide semiconductors (CMOS) [4]. With the development of micromachining technology, various MEMS Pirani vacuum gauges with a complex structure and small size have been designed and fabricated [4].

A lot of research has been devoted to expanding the measurement range of MEMS Pirani vacuum gauges and reducing the size of the sensor. Table 1 summarizes such progress in the last two decades. In these 20 years of research, most of the Pirani vacuum gauges prepared by researchers have lower sensitivity at higher vacuum and near atmospheric pressure, which is caused by the blind pursuit of expanding the dynamic range of Pirani vacuum gauges. In reference [5], two different sizes of Pirani gauges were prepared to achieve a low-pressure range and high-pressure range, but the individual ranges were small and the microbridge structure was long with poor mechanical properties. The Pirani gauge in reference [6] has a large measuring range but low sensitivity in the low- and high-pressure range, which reduces the practical performance. In addition, the Pirani gauges in references [1,2,3,7,8] have generally smaller ranges. Furthermore, the Pirani gauges in references [4,9,10,11,12,13,14] have problems relating to complex structure, difficult manufacturing processes, and high cost. The Pirani gauges in references [4,15,16] have a similar dynamic range as those in our work, but reference [4] uses ion implantation to form a diode as the thermal layer, which greatly increases the complexity of the process. In contrast, our thermal layer needs only to be patterned after depositing a thin vanadium oxide film using a PVD process, which improves the stability of the device while reducing the process complexity. Although the dynamic range of the Pirani gauges in references [15,16] is large, the Pirani meter in reference [15] undergoes a large reduction in sensitivity when the pressure is greater than 1333 Pa, while the Pirani meter in reference [16] is significantly less sensitive in both range segments where the pressure is less than 1 Pa and where the pressure is greater than 1000 Pa. Therefore, our results are superior in these two range bands. In addition to large dynamic vacuum pressure range and small sensor size, much attention has been focused on sensitivity and power consumption due to the application of the Pirani vacuum gauges in portable measuring equipment [4]. In order to achieve high sensitivity in Pirani vacuum gauges, and in order to be suitable for general vacuum MEMS devices, especially for direct integration with VOx-based uncooled infrared sensors, we propose a novel Pirani vacuum gauge.

The novel Pirani vacuum gauge we proposed uses a VOx membrane as the thermosensitive layer to achieve high sensitivity at high vacuum. The high temperature coefficient of resistance (TCR) of VOx increased the sensitivity of the Pirani vacuum gauge, while low thermal conductivity reduced the detection limit [7]. In addition, we optimized the floating structure to decrease the thermal conductivity so that the detecting range of the Pirani gauge tended to be in the low-pressure end. The Pirani vacuum gauge made using a VOx membrane is compatible with CMOS and can be easily integrated into VOx-based uncooled infrared (IR) FPAs to monitor the pressure and modify the output. This work is related to the design stage only of the device, while the fabrication and characterization of the device will be reported in the future.

## 2. Principle and Heat Transfer

The heat transfer mechanism of the simple Pirani gauge is shown in Figure 1. The Pirani gauge consists of a resistance heater (i.e., a thermal sensitive layer) and a thermal insulating layer to form a suspended structure. At a certain pressure, the heat generated by the resistance heater ($Q_{total}$) and the heat dissipated reach an equilibrium, so that the resistance heater is stable at a certain temperature. Typically, the total heat losses consist of three parts [4,19]:
Solid conduction from cantilever beam to its carrier ($Q_{solid}$); the value of the thermal conductance depends on the geometry of the sensor’s carrier and its thermal conductivity.Radiation from the sensor’s hot surface to the surface of the chamber ($Q_{rad}$); its value depends on the emissivity and the exterior surface of the sensor.Gas convection and conduction ($Q_{gas}$); this effectively depends on pressure.


The equilibrium equation can be expressed as:(1)$$Q_{total}=Q_{solid}+Q_{rad}+Q_{gas}\left(P\right),$$
(2)$$Q_{total}=\left(T-T_{0}\right)\left(G_{solid}+G_{rad}+G_{gas}\left(P\right)\right),$$
where *T* is the equilibrium temperature, $T_{0}$ is the initial temperature, $G_{solid}$ is the solid thermal conductance of the sensor carrier, $G_{rad}$ is the radiation thermal conductance, and $G_{gas}\left(P\right)$ is the gas thermal conductance. Since the heat dissipated by gas thermal conductance is related to pressure, while the heat dissipated by solid conduction and radiation is independent of pressure, the total heat dissipated also changes when pressure changes. This means that the Pirani gauge corresponds to different thermal sensitive layer temperatures at different pressures. Since the resistance of the thermal sensitive layer changes with temperature, the resistance of the thermal sensitive layer is affected by the ambient pressure of the Pirani gauge, and the ambient pressure can be reflected by measuring the resistance of the thermal sensitive layer of the Pirani gauge [5].

### 2.1. Solid Conduction

Solid conduction occurs in the continuum materials where a temperature gradient exists [2]. Thermal conductivity of a homogeneous solid material remains constant under a certain temperature, which is independent of the variation of gas pressure [3]. The thermal conductance by the solid can be evaluated according to Fourier’s Law of heat conduction [7,18,19,20]:(3)$$G_{solid}=4\lambda_{solid}\frac{A_{s}}{L_{s}},$$
where $\lambda_{solid}$ is the thermal conductivity of the cantilever beam, $A_{s}$ is the cross-sectional area of the contact position of the cantilever beam and the sensor wall, and $L_{s}$ is the length of the cantilever beam. Since there are four cantilevers, the thermal conductance by the solid should be multiplied by four.

### 2.2. Radiation

Radiation can be calculated by Stefan-Boltzmann’s law [18,19]:(4)$$Q_{rad}=2\varepsilon A\sigma_{b}\left(T^{4}-T_{0}^{4}\right),$$
(5)$$G_{rad}=\frac{Q_{rad}}{T-T_{0}}=2\varepsilon A\sigma_{b}\left(T+T_{0}\right)\left(T^{2}+T_{0}^{2}\right),$$
where *ε* is the surface emissivity, *A* is the hot-plate area, and $\sigma_{b}$ is the Stefan-Boltzmann constant. Because there are two surfaces on the microplate, the thermal radiation value is doubled.

### 2.3. Gas Convection and Conduction

Heat dissipation through the gas consists of two parts: thermal convection and conduction. Gas convection heat transfer refers to the heat transfer generated by gas through the solid surface at macro scale, while gas conduction heat dissipation refers to the heat exchange caused by the collision between gas molecules in the chamber and the sensor wall at micro scale [2]. They are related to the Knudsen number ($K_{n}$), which is defined as:(6)$$K_{n}=\frac{l}{d},$$
where *l* is the mean free path of the gas, and *d* is the characteristic dimension of the chamber (air gap).

According to the gas molecular dynamics theory, the gas heat transfer model is related to the value of $K_{n}$: when $K_{n}<0.01$, the gas can be regarded as a continuum; when $K_{n}>10$, the gas is regarded as a free molecular flow; when $0.01<K_{n}<10$, the gas can be further divided into turbulent flow and transition flow. Turbulent flow means $0.01<K_{n}<0.1$, and transition flow means $0.1<K_{n}<10$. If *d* = 5 μm in the prepared Pirani gauge when the pressure range is from 0.01 Pa to 10^4^ Pa, $K_{n}>0.01$, the gas is in free molecular flow and transition flow. Therefore, in the gas range studied, gas convection does not occur, as the continuum would fail in this case. Hence, the heat dissipated through the gas is dissipated only by gas conduction.

In this case, the thermal conductance of the gas needs to be calculated using the rarefied gas heat transfer theory and the extended Fourier law [18]:(7)$$G_{gas}\left(P\right)=\lambda \left(P,d_{1}\right)\frac{A}{d_{1}}+\lambda \left(P,d_{2}\right)\frac{A}{d_{2}},$$
where *d* is the air gap, $d_{1}$ is the distance between the lower surface of the hot plate and the bottom surface of the cavity, and $d_{2}$ is the distance between the upper surface of the hot plate and the package cover, and $\lambda \left(P,d\right)$ is the thermal conductivity of the gas when the pressure is *P* and the air gap is *d* [18,19]. Furthermore:(8)$$\lambda \left(P,d\right)=\lambda \left(P_{0}\right){\left(\;\left(\frac{2-a}{a}\right)\frac{l}{d}\frac{9.5}{6}\right)}^{-1},$$
where $\lambda \left(P_{0}\right)$ is the thermal conductivity of the gas at atmospheric pressure, $\lambda \left(P_{0}\right)=25.6\times 10^{-3}W/\left(m\cdot K\right)$; a is the energy adjustment coefficient of the gas molecule between the two surfaces, for $N_{2}$ in the chamber, a = 0.77; and *l* is the mean free path of gas molecules, $l\left(P\right)\cdot P=6.67\times 10^{-3}m\cdot \mathrm{Pa}$.

## 3. Design and Simulation

### 3.1. Material

According to the operating principle of the Pirani gauge, the smaller the ratio of solid conduction and radiation (the proportion of $Q_{gas}\left(P\right)$ in Equation (1) is larger), the greater the influence of pressure on the equilibrium of Equation (1). In order to extend the measurement range to lower-pressure states, a larger $Q_{gas}\left(P\right)/Q_{solid}$ must be achieved [5], so the thermal conductivity of the support material should be minimized. The commonly used support materials are ${\mathrm{Si}}_{3}N_{4}$ and ${\mathrm{SiO}}_{2}$, with a thermal conductivity of $20\left[W/\left(m\cdot K\right)\right]$ and $1.4\left[W/\left(m\cdot K\right)\right]$, respectively. Therefore, ${\mathrm{SiO}}_{2}$ was chosen as the support layer and our choice was verified by simulation. The TCR of platinum (Pt), titanium (Ti), and other metals commonly used in Pirani gauge is $0.3{\%K}^{-1}∼0.4{\%K}^{-1}$, while the TCR of VOx can reach $-3{\%K}^{-1}$, which is one order of magnitude higher than that of metals. This means that if the Pirani gauge is made with VOx, its resistance changes more significantly at the same temperature change, resulting in higher sensitivity. Therefore, we chose VOx as the thermal sensitive layer of the Pirani gauge. For the electrode material, aluminum (Al), which is most used in semiconductor manufacturing process, was selected.

### 3.2. VOx TCR Measurement

Since the Pirani gauge we designed needed to utilize the conductivity of VOx films as a function of temperature, we designed experiments and tested the TCR of VOx films. For the preparation of VOx films, we first deposited a 700 nm thick silicon nitride (${\mathrm{Si}}_{3}N_{4}$) film on a bare wafer, then a 60 nm thick VOx film, and finally annealed at 280 °C for 5 min. Since the curves of VOx film sheet resistance versus temperature do not coincide in the warming and cooling stages, there is a relationship similar to the hysteresis curve, but the VOx properties can be stabilized by annealing, while the two curves are nearly coincident. We prepared the VOx films by DC magnetron sputtering and pre-deposited two dummy wafers to bring the machine to a steady state before preparing the experimental wafers. Argon plasma played the main role of bombardment, while oxygen plasma reacted with the bombarded vanadium atoms on the wafer surface to form VOx. We then measured the sheet resistance of VOx films at different temperatures using a four-probe stage with a heating function. We measured the sheet resistance for VOx between 23 °C and 70 °C at 1 °C intervals. Finally, the data were analyzed to obtain the relationship between temperature and VOx sheet resistance, as shown in Figure 2a, where the sheet resistance of the VOx film is exponentially related to temperature. After that, we took the logarithm of the sheet resistance of the VOx film and then fitted it to the temperature, and found that the linearity of the fit was extremely high, thus verifying our conjecture, as shown in Figure 2b.

### 3.3. Structure and Readout Circuit Design

As shown in Figure 3, we designed a high-sensitivity Pirani gauge based on VOx film for high vacuum measurements. A cavity was formed on the substrate and four cantilever beams were used to suspend the VOx film above the cavity. The first layer above the cavity was a support layer containing four cantilevers, the second layer was a thermal sensitive layer, and the top layer was a protective layer and two electrodes. The design is simple in structure, easy to manufacture, and compatible with conventional semiconductor manufacturing process. At the same time, this Pirani gauge structure is consistent with the VOx-based uncooled IR imaging sensor structure and can be integrated directly inside the uncooled IR sensor. We finalized the material and dimensional parameters of each layer through extensive simulation experiments. The support layer of our Pirani gauge was silicon dioxide (${\mathrm{SiO}}_{2}$), the protective layer was silicon nitride (${\mathrm{Si}}_{3}N_{4}$), and the electrode was aluminum (Al). Our basic model dimensions included the thickness of support layer, thermal sensitive layer, protective layer and electrodes, which were 300 nm, 110 nm, 80 nm, and 150 nm, respectively. The central thermal sensitive area was 130 × 130 μm^2^, the width of the cantilever beam was 13μm, and the depth of the cavity was 5 μm. In addition, we opened some holes in the central thermal area, the purpose of which was twofold. On the one hand, our preparation process started with etching the cavity, followed by filling the polyimide (PI) as a sacrificial layer, and finally releasing the PI. Since the lateral release width of PI was limited, we discovered that if we used the release window at the edge of the cantilever beam, the PI below the thermal zone in the cavity could not be completely released, so we opened some release holes above the thermal zone. On the other hand, since the Pirani gauge generates thermal stress when heated, if the stress was too large to cause the cantilever to break, the stress was released through these holes.

The conventional Pirani gauge thermal layer is mostly a meandering metal wire, which aims to heat the whole thermal zone uniformly. However, we used the whole VOx film as the thermal layer, mainly because the VOx film has a larger resistance compared to the metal. We conducted simulations for the meandering VOx film and the whole VOx film, respectively. When using the meandered VOx film, the temperature of VOx reached about 70 °C with the addition of 15 V at both ends of the Pirani gauge, while it reached about 70 °C with the addition of only 6.5 V when using the whole VOx film. Using meandering VOx film increased power consumption without improving performance, so we used the whole VOx film as the thermal layer material.

In order to characterize the change in resistance of the Pirani gauge’s thermal sensitive layer, a Wheatstone bridge circuit was used and modified, as shown in Figure 4 [6]. In the figure, RS is the Pirani gauge and RR is the reference resistor, where the reference resistor was also used with VOx (the same structure as Pirani gauge, except with no cavity underneath) and placed next to the Pirani gauge, whose purpose was to eliminate the effect of ambient temperature. The resistance of R1, R2, and RL was 5 kΩ, 5 kΩ, and 1 MΩ, respectively. The change in pressure in the vacuum cavity was characterized by detecting the change in RL after a constant voltage was applied.

### 3.4. Fabrication Process Design

Figure 5 demonstrates the fabrication process of the novel Pirani gauge. First, a cavity was etched on the surface of the bare wafer as shown in Figure 5a. Then, spin-coated polyimide (PI) was used as a sacrificial layer to fill the cavity, as shown in Figure 5b. Since the upper surface of spin-coated PI was not flat, we then flattened the surface using chemical mechanical polishing (CMP), as shown in Figure 5c. After that, ${\mathrm{SiO}}_{2}$, Vox, and ${\mathrm{Si}}_{3}N_{4}$ were deposited as the support layer, thermal sensitive layer, and protective layer, respectively, as shown in Figure 5d. Immediately afterwards, ${\mathrm{Si}}_{3}N_{4}$ and VOx were patterned as shown in Figure 5e. Then, ${\mathrm{Si}}_{3}N_{4}$ on the cantilever was etched out of the window for electrode connection, as shown in Figure 5f. Next, Al was deposited and etched to form the Al electrodes as shown in Figure 5g,h. Then, ${\mathrm{SiO}}_{2}$ was etched to create a window for subsequent sacrificial layer release, as shown in Figure 5i. The final step was to release the PI in the Asher machine using oxygen plasma as shown in Figure 5j. Since steps (e,i) completed the etching of the release hole as well as the release window at the edge of the cantilever beam, the lateral release width of the PI was greatly reduced in the last step so that the PI in the cavity could be completely released. This whole process flow was the same as the preparation process of a VOx-based uncooled IR imaging sensor, so this Pirani gauge can be manufactured simultaneously with the uncooled IR sensor to achieve wafer-level packaging.

### 3.5. Simulation of Pirani Gauge

We used COMSOL Multiphysics to simulate the Pirani gauge. We deduced the conductivity versus temperature from the VOx film resistance versus temperature and entered it into the VOx material parameters in the software. We added “Joule Heat” and “Heat Transfer in Solids” physical fields to the software, and added surface-to-ambient radiation to the “Heat Transfer in Solids” physical field. A temperature boundary condition of 293[K] was set at the bottom of the device. In addition, heat flux was set on the lower surface of the support layer and the upper surface of the protection layer to replace the gas thermal conduction. The heat flux calculation formula is as follows:(9)$$q=\frac{\lambda \left(P,d\right)}{d}\left(T_{0}-T\right),$$
where q is the heat flux, and $\lambda \left(P,d\right)$ is the thermal conductivity of the gas when the pressure is *P* and the air gap is *d*.

## 4. Optimization of Pirani Gauge Parameters

We simulated the material and thickness of the support layer, the area of the thermal sensitive zone, the width of the cantilever beam, and the air gap. We analyzed the simulation results and used them to optimize the device we designed. Finally, we designed a high-sensitivity Pirani gauge based on a thin vanadium oxide film for high vacuum measurement.

### 4.1. Support Layer Materials

As shown in Figure 6, when the support layers were ${\mathrm{Si}}_{3}N_{4}$ and ${\mathrm{SiO}}_{2}$, we found that ${\mathrm{SiO}}_{2}$ performed better in the same thickness of the support layers. This is mainly because the thermal conductivity of ${\mathrm{Si}}_{3}N_{4}$ ($20\left[W/\left(m\cdot K\right)\right]$) is greater than that of ${\mathrm{SiO}}_{2}$ ($1.4\left[W/\left(m\cdot K\right)\right]$), leading to an increase in the proportion of solid thermal conduction and a decrease in the proportion of gas thermal conduction, which ultimately leads to a decrease in the temperature reached by the thermosensitive layer. In addition, the large amount of heat loss through solid conduction leads to a smaller proportion of gas conduction, resulting in a lower sensitivity of the Pirani gauge.

### 4.2. Thickness

The simulation results for the thickness of the support layer are shown in Figure 7. As shown in Figure 7a,b, when the support layer was ${\mathrm{SiO}}_{2}$ and the voltage and VOx thickness were constant, as the ${\mathrm{SiO}}_{2}$ thickness increased, the heat absorbed by ${\mathrm{SiO}}_{2}$ increased and the maximum temperature that VOx could reach decreased, while the proportion of heat dissipated by solid conduction increased, which led to a decrease in the sensitivity of the Pirani gauge in the low-voltage range. The sensitivity of the Pirani gauge increased slightly in the low-pressure range when the thickness of ${\mathrm{SiO}}_{2}$ was reduced from 300 nm to 250 nm, but there was a greater impact on the mechanical structure of the Pirani gauge. Since the ${\mathrm{SiO}}_{2}$ needed to support the VOx and ${\mathrm{Si}}_{3}N_{4}$ above, we chose a 300 nm thick ${\mathrm{SiO}}_{2}$. As shown in Figure 7c,d, as the thickness of the VOx increased, the resistance of the VOx decreased, thus generating more heat and therefore allowing the VOx to reach higher temperatures. When the thickness of VOx increased from 110 nm to 120 nm, the temperature had a large increase, but the range and sensitivity of the Pirani gauge did not change significantly, and the thermal stress generated when the temperature was higher was also larger, which was not conducive to the mechanical structure of the Pirani gauge, so we chose a thickness of 110 nm for the VOx film.

### 4.3. Thermal Sensitive Area

As shown in Figure 8a,b, through the study of the simulation of the thermal zone area, we found that with the increase of the thermal sensitive area, the maximum temperature value that the thermal sensitive layer can reach increased and the sensitivity of the Pirani gauge increased. This is mainly because after the thermal zone area was increased, gas molecules and the hot plate contacted more fully, resulting in a gas conduction increase, which also increased the sensitivity of the Pirani gauge under high vacuum conditions. We have simulated the sensitivity of the Pirani gauge for different pressure interval segments separately, and the sensitivity of the Pirani gauge shows the change, as shown in Figure 8c,d, with the increase of the area of the thermal zone. The sensitivity *S* of the Pirani gauge is defined as the slope of the device output voltage variation curve with a logarithmic air pressure of $\log_{10}P$ [21]. It is calculated as follows:(10)$$S=\frac{\partial V}{\partial \left(\log_{10}P\right)},$$
where *S* is the Pirani gauge sensitivity, and *V* is the device output voltage. As the area of the thermal zone increased, the sensitivity of the Pirani gauge in the low-pressure range gradually increased, while the sensitivity in the high-pressure range gradually decreased. When the overall size of the device was determined (i.e., the cavity area was determined), we found a good thermal area of 130 × 130 μm^2^.

### 4.4. The Depth of the Cavity

As shown in Figure 9, after simulation, it was found that the air gap had no obvious influence on the Pirani gauge performance. This is because we added two heat fluxes to replace the conduction of gas molecules in the simulation, as shown in formula (9). We substituted Equation (8) into Equation (9), after simplification:(11)$$q=\frac{\lambda \left(P_{0}\right)}{d+5l}\left(T-T_{0}\right),$$
where *l* is the mean free path of a gas molecule. When pressure *P* < 1000 *Pa*, *d* ≪ 5 *l*, the effect of *d* on q can be ignored; when the pressure *P* > 1000 Pa, *d* begins to affect q, and as the pressure continues to increase, the influence of *d* becomes more obvious.

## 5. Conclusions

We tested the sheet resistance of the VOx film as a function of temperature and used it in simulation software to determine the electrical performance of our designed Pirani gauge. We determined that the best results were when the support material was ${\mathrm{SiO}}_{2}$ through simulation of the support material type. In addition, we have simulated the structure size and thickness of each layer of the Pirani gauge to finalize our Pirani gauge. We also designed and simulated the readout circuit and designed the fabrication process to be compatible with the VOx-based uncooled IR imaging sensor process so that wafer-level packaging can be achieved. Finally, we designed a miniature hot-plate Pirani vacuum gauge with a simple structure and compatibility with conventional semiconductor manufacturing processes. This Pirani gauge has a cavity depth of 5 μm, a support layer made of ${\mathrm{SiO}}_{2}$ with a thickness of 300 nm, and a thermal sensitive layer of VOx with a thickness of 110 nm. The material of the two electrodes is Al with a thickness of 150 nm. The central thermal sensitive area is 130 × 130 μm^2^ and the cantilever beam width is 13 μm. It has a range of 10^−1^~10^4^ Pa, an average sensitivity over the whole range of 1.23 V/lgPa, and an average sensitivity in the range of 1~1000 Pa is 1.87 V/lgPa. These concluding data are all our design simulation data, while the actual characterization work and related experimental results will be reported in the future.

## Figures and Tables

**Figure 1 sensors-22-09275-f001:**
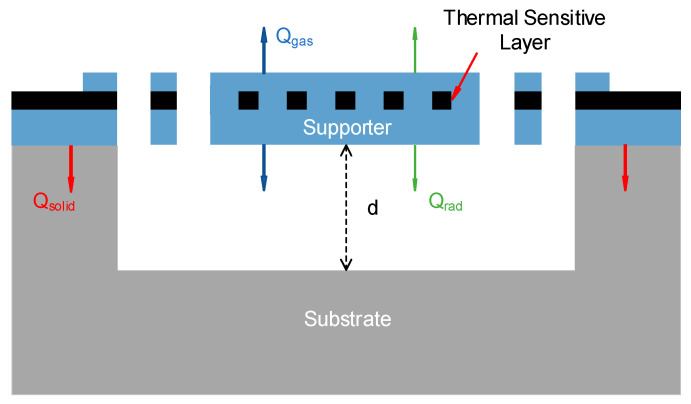
Heat transfer mechanism of the Pirani gauge.

**Figure 2 sensors-22-09275-f002:**
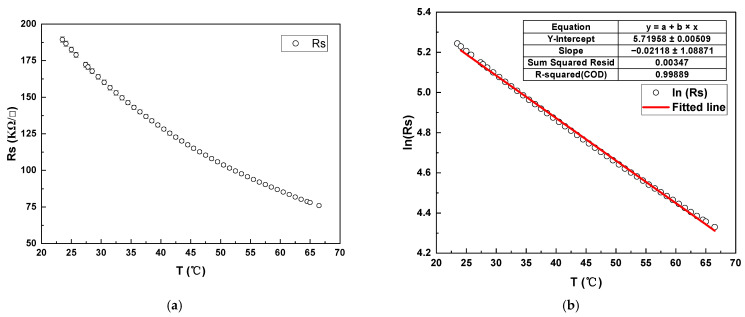
(**a**) Sheet resistance values of VOx films at different temperatures; (**b**) Linearity of ln(Rs) of VOx films versus temperature T.

**Figure 3 sensors-22-09275-f003:**
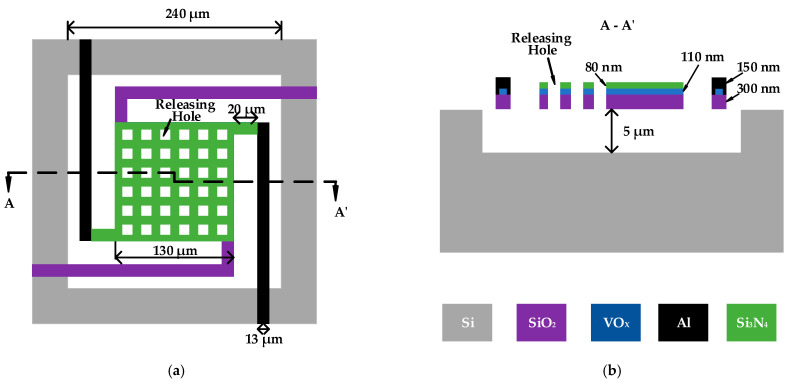
Schematic diagram of the Pirani gauge. (**a**) Top view of the Pirani gauge; (**b**) AA′ cross-section view of the Pirani gauge.

**Figure 4 sensors-22-09275-f004:**
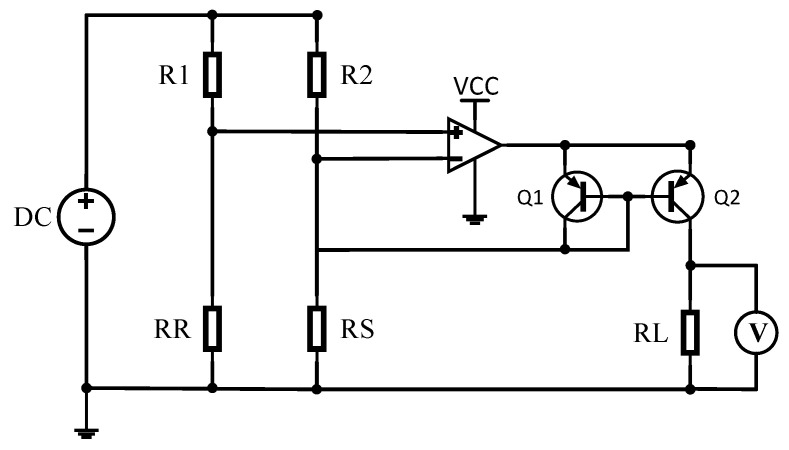
Schematic diagram of Pirani gauge readout circuit.

**Figure 5 sensors-22-09275-f005:**
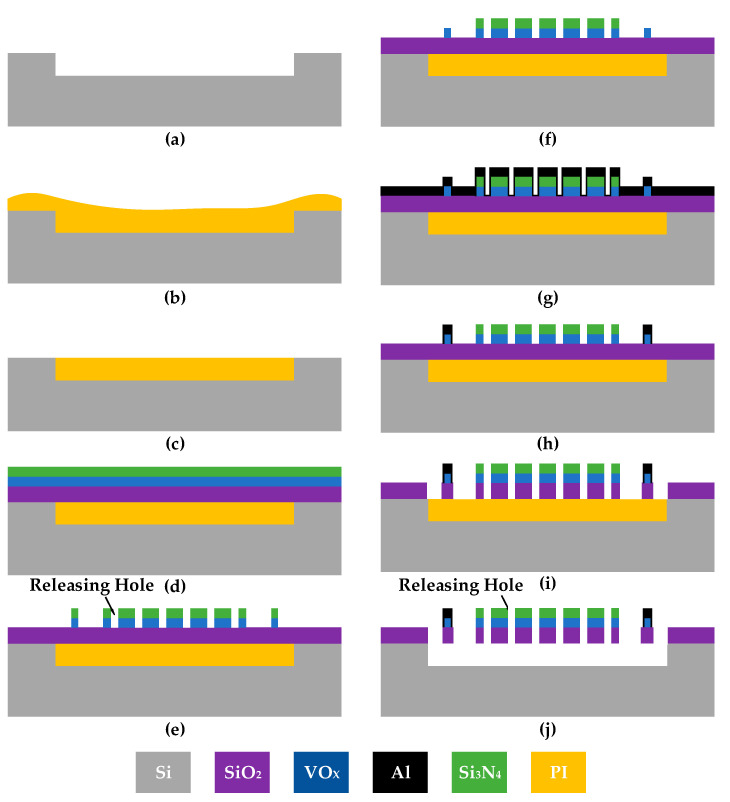
Fabrication flow diagram of VOx Pirani gauge. (**a**) Cavity Etch. (**b**) PI coating and curing. (**c**) CMP flattening. (**d**) SiO_2_, VOx, Si_3_N_4_ thin film deposition. (**e**) VOx and Si3N4 etching. (**f**) Electrode Window Etch. (**g**) Al thin film deposition. (**h**) Al Electrode Etch. (**i**) PI Release Window Etch. (**j**) PI Release.

**Figure 6 sensors-22-09275-f006:**
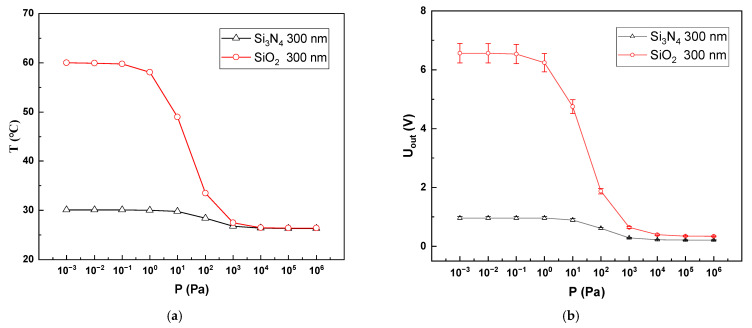
Ranges of Pirani gauges at different support materials. (**a**) Trend of thermal layer temperature with pressure for different support layer materials; (**b**) Trend of Pirani gauge output voltage with pressure for different support layer materials.

**Figure 7 sensors-22-09275-f007:**
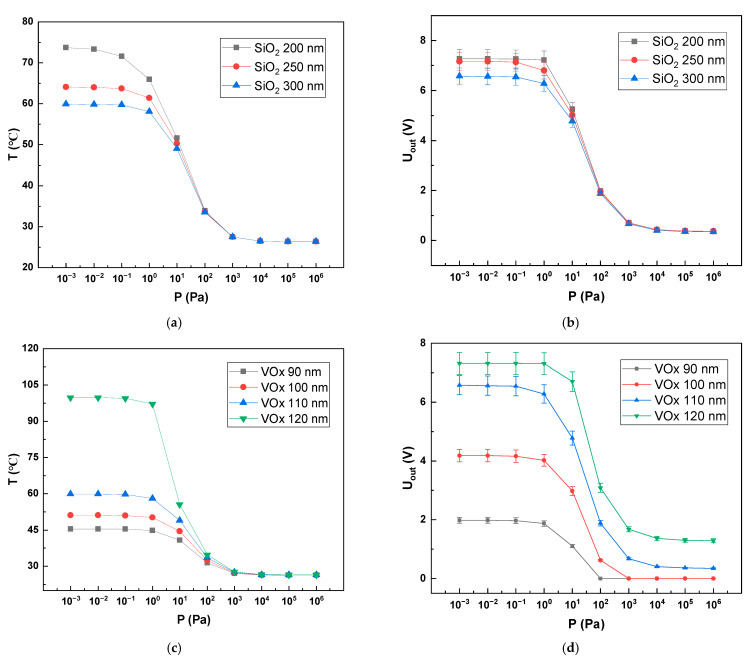
VOx temperature and output voltage for different film thicknesses. (**a**) Trend of VOx film temperature with pressure for Pirani gauge with different ${\mathrm{SiO}}_{2}$ thicknesses; (**b**) Trend of output voltage with temperature for devices with different ${\mathrm{SiO}}_{2}$ thicknesses; (**c**) Trend of thermal layer temperature with pressure for devices with different VOx thicknesses; (**d**) Trend of output voltage with pressure for devices with different VOx thicknesses.

**Figure 8 sensors-22-09275-f008:**
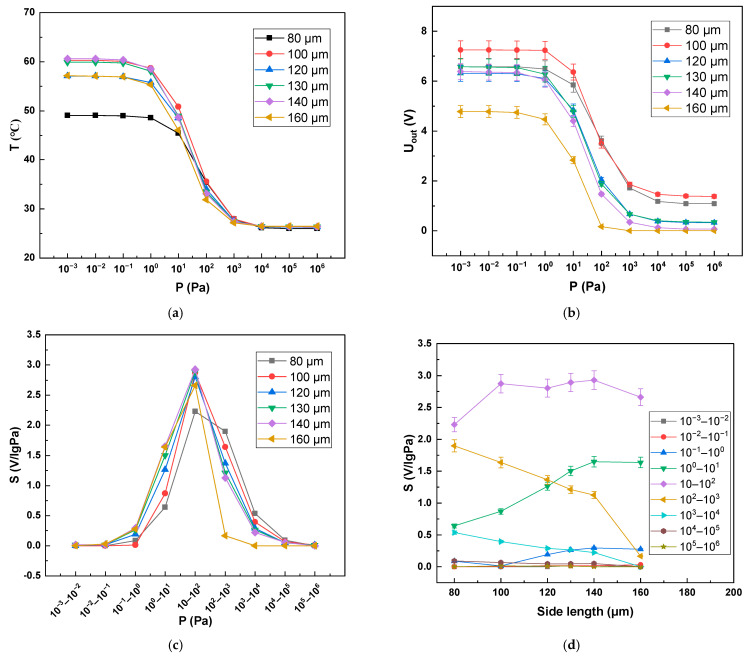
Range and sensitivity of Pirani gauge in different heat sensitive areas. (**a**) Variation trend of VOx temperature with pressure in Pirani equilibrium state under different thermal sensitive areas; (**b**) Variation trend of output voltage with pressure in Pirani equilibrium state under different thermal sensitive areas; (**c**) Sensitivity of Pirani gauge at different pressure intervals for different thermal sensitive areas; (**d**) The influence of the area of thermal sensitive zone on the sensitivity of Pirani gauge in different pressure ranges.

**Figure 9 sensors-22-09275-f009:**
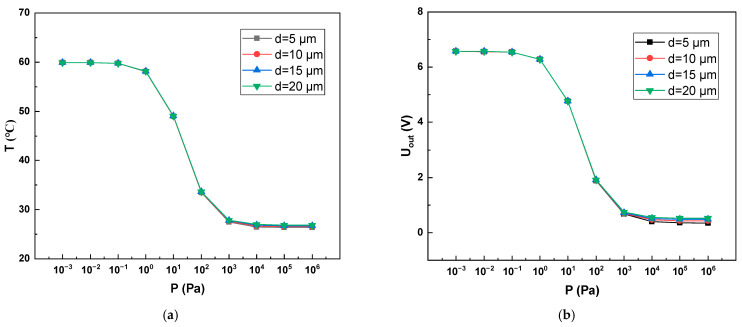
Range of Pirani gauge at different cavity depths. (**a**) Variation of Pirani gauge VOx temperature with pressure at different cavity depths; (**b**) Variation of Pirani gauge output voltage with pressure at different cavity depths.

**Table 1 sensors-22-09275-t001:** Detection principles and pressure ranges of MEMS Pirani gauges.

Researcher	Type of Gauge	Pressure Range (Pa)
Mitchell, J., et al. (2008) [5]	Microbridge	0.1~1 × 10^3^ 6.7~1 × 10^5^
Chen, Y.-C., et al. (2015) [17]	Microbridge	6.7~1.3 × 10^4^
Wang, S., et al. (2001) [6]	Resistor on dielectric membrane (Ti)	7 × 10^−3^~1 × 10^5^
Chae, J., et al. (2005) [9]	Resistor on dielectric membrane (Cr/Pt)	2.7~266.6
	Lateral heat transfer (P++ silicon)	
Dong, T., et al. (2009) [2]	Resistor on dielectric membrane (Si/Ge)	0.1~100
Wang, X., et al. (2010) [1]	Resistor on dielectric membrane (Pt)	1~300
Völklein, F., et al. (2013) [18]	Resistor on dielectric membrane (Ni)	1.3 × 10^−4^~133
Grau, M., et al. (2015) [15]	Resistor on dielectric membrane (Ni)	0.1~1 × 10^5^
Lecler, S., et al. (2019) [16]	Resistor on dielectric membrane (Ni)	10^−2^~10^5^
Wei, D., et al. (2019) [4]	Resistor on dielectric membrane (Series diode)	10^−1^~10^4^
Liu, C., et al. (2020) [10]	Resistor on dielectric membrane (Series diode)	1~500
Topalli, E. S., et al. (2009) [8]	P++silicon coil microbridge 100μm thick silicon coil microbridge	1.3~266.6 6.7~666.6
Kim, G., et al. (2014) [11]	Resistor on dielectric membrane (boron-doped α-Si)	13.3~1333
Sun, Y.-C., et al. (2015) [12]	Heater-with-holes	27~2.7 × 10^4^
Kong, Y., et al. (2017) [13]	Comb-shape structure	0.5~2.6 × 10^3^
Mo, J., et al. (2020) [14]	Suspension of poly-SiC membrane	10~10^5^
Xiao, B., et al. (2011) [3]	Resistor on dielectric membrane (VOx)	1~100
Garg, M., et al. (2021) [7]	Suspension of V_2_O_5_ membrane	4~2.7 × 10^3^
This Work	Resistor on dielectric membrane (VOx)	10^−1^~10^4^

## Data Availability

Not applicable.

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
