# Peer review of "Design of a High Sensitivity Pirani Gauge Based on Vanadium Oxide Film for High Vacuum Measurement"

_sensors, 2022, doi:10.3390/s22239275_

Round 1

Reviewer 1 Report

The paper presents designs for a novel Pirani gauge. A fabrication process is proposed, and an optimized design is developed. The presented work is interesting and thorough; however, the lack of an experimental demonstration is disappointing. Even without an experimental demonstration, the work presented in this paper is interesting and merits publication.

The paper is recommended for publication after the following minor revisions:

1)      Line 38 randomly has the word “seriously” in it.

2)      Layer thicknesses should be included in Figure 2.

3)      Is Figure 7 simulation? Needs to be more clear.

4)      It makes more sense for section 3.3 and 4.1 to be earlier in the paper. At the beginning of section 3 would make more sense. It seems weird that the discussion of why VOx is used (section 4.1) is after the section on VOx measurements (section 3.3). And both of these sections should come before the discussion of the sensor structure in section 3.1.

5)      The paper as it is written, is difficult to understand that it is predominantly a simulation paper. I would make it more clear in the text that most of the results presented here are simulated, and that the proposed fabrication process and circuit are designed but not implemented. Reordering the paper by presenting sections 3.3 and 4.1 earlier in the paper would also help make this more clear.

Author Response

Point 1: Line 38 randomly has the word “seriously” in it.

Response 1: In line 38, I deleted the word "seriously".

Point 2: Layer thicknesses should be included in Figure 2.

Response 2: I have marked the thickness of each layer of the film in Figure 2(b).

Point 3: Is Figure 7 simulation? Needs to be more clear.

Response 3: Figure 7 shows the simulation results. To make the presentation clearer, I added " The simulation results for the thickness of the support layer are shown in Figure 7." at the beginning of this section.

Point 4: It makes more sense for section 3.3 and 4.1 to be earlier in the paper. At the beginning of section 3 would make more sense. It seems weird that the discussion of why VOx is used (section 4.1) is after the section on VOx measurements (section 3.3). And both of these sections should come before the discussion of the sensor structure in section 3.1.

Response 4: For sections 3.3 and 4.1 in the paper, the following adjustments have been made. I have moved the selection of materials in section 4.1 and section 3.3 to the first and second subsections of section 3, respectively.

Point 5: The paper as it is written, is difficult to understand that it is predominantly a simulation paper. I would make it more clear in the text that most of the results presented here are simulated, and that the proposed fabrication process and circuit are designed but not implemented. Reordering the paper by presenting sections 3.3 and 4.1 earlier in the paper would also help make this more clear.

Response 5: For sections 3.3 and 4.1 in the paper, the following adjustments have been made. I have moved the selection of materials in section 4.1 and section 3.3 to the first and second subsections of section 3, respectively.

Author Response

Point 1: Authors provided a comparison of their work to previous works in Table 1. Authors should sort the comparison based on the type of gauge and pressure range in logical order. Thus readers can easily identify the better one. It will be interesting if authors could compare their work to works in ref [4, 13, 16] which have similar dynamic ranges.

Response 1: I modified Table 1 to categorize the Pirani gauges listed in the table for comparison according to the type of gauge, and added a comparison of our work with that of references [4, 13, 16] with similar dynamic ranges. The modifications are shown below (The references corresponding to the original references [4, 13, 16] became [4, 15, 16] due to the reclassification of the Pirani gauge tables listed in the table.):

References [4, 15, 16] have a similar dynamic range as our work, but reference [4] uses ion implantation to form a diode as the thermal layer, which greatly increases the complexity of the process. In contrast, our thermal layer only needs to be patterned after depositing a thin vanadium oxide film using a PVD process, which improves the stability of the device while reducing the process complexity. Although the dynamic range in references [15, 16] is large, the Pirani meter in reference [15] undergoes a large reduction in sensitivity when the pressure is greater than 1333Pa, while the Pirani meter in reference [16] is significantly less sensitive in both range segments where the pressure is less than 1Pa and where the pressure is greater than 1000Pa. And our work is significantly better than them in these two range bands.

Point 2: There is a lack of device and materials characterization. Because the device's performance heavily depends on its structure, it is good to provide the actual structure of the device after its fabricated and focus on its structure not only on schematic illustration. Each of the materials must be characterized. How to control the thickness of the layers? and the measurement of the layers should be provided.

Response 2: Since our devices are not fully fabricated yet and are still under fabrication, we cannot provide the actual fabricated structures. We monitor the thickness of the film by using an ellipsometer to measure the thickness of silicon dioxide and silicon nitride. For the thickness of the aluminum layer, we measure the square resistance of the aluminum using a four-probe stage and calculate the thickness of the aluminum layer by the relationship between resistivity and square resistance. For the thickness of vanadium oxide film, we use a four-probe bench to measure the square resistance after deposition. For the thickness of the film, we slice the deposited VOx film and then use SEM to measure the film thickness, and when our machine recipe is not adjusted, we approximate the thickness of VOx on our product to be equal to the thickness measured by the previous slice.

Point 3: Generally, there are no error bars for every data in the graphs. The error bars should be provided.

Response 3: Since our devices are not yet finished being manufactured, the data in the graphs are simulated data, so we do not have error statistics for the test data versus the real data for now, so we do not provide error bars in our graphs.

Round 2

Reviewer 2 Report

Dear Authors, 
After critically analyzing the responses from the authors to my comments, I found that important data were missing which makes this manuscript not ready to publish in the sensors in this current version. Authors need more time to do experiments to generate sufficient data. 

Best Regard, 

Author Response

Thank you very much for taking the time to give us your valuable review comments. Since my previous response was not well prepared, I will again make an addition.
